# Evaluation of Digital Transformation to Support Carbon Neutralization and Green Sustainable Development Based on the Vision of "Channel Computing Resources from the East to the West"

**Zhaoyang Wu** [1,2,3]**, Xiaoning Wang** [1]**, James Yong Liao** [2,*]**, Hongrong Hou** [2] **and Xiaokui Zhao** [1]

[1] School of Economics and Management, Qinghai Normal University, Xi'ning 810016, China
[2] Center of International Education, Philippine Christian University, Malate, Manila 1004, Philippines
[3] The Center of Provincial-Ministerial Collaborative Innovation for Research on Tibetan History and Multi-Ethnic Prosperity and Development, Xi'ning 810016, China
* Correspondence: james.liaoyong@pcu.edu.ph

**Abstract:** The long-term dependence on fossil fuels has led to an increase in carbon dioxide emissions. Global warming poses a huge risk to the sustainable development of the world, and even threatens human survival. The arrival of the carbon neutral era means that urban development is facing serious restrictions on carbon emissions. Digitization has brought profound changes to the economic and social development model, and would also change the pattern of urban competition. The goal of carbon neutrality is to change the low-carbon development model and structure, supplement it with negative carbon emissions, and comprehensively reduce greenhouse gas emissions. However, achieving the goal of carbon neutrality still faces many challenges and problems. For this reason, this paper analyzed the significance of carbon neutralization and the challenges faced by sustainable development to study the advantages of carbon neutralization under Digital Transformation (abbreviated as DT), and finally proposed the implementation path of carbon neutralization and sustainable development based on the channel of computing resources from the east to the west. The carbon emission effect before DT increased with time, while the carbon emission effect after DT decreased with time, in which the carbon emission effect after DT decreased by 47.9% compared with that before DT. The post-DT industry digitalization degree and the carbon trading system perfection degree were better than those before DT. The post-DT industry digitalization degree was 10.4% higher than that before DT, and the carbon trading system perfection degree was 9.5% higher than that before DT. In a word, DT and channeling computing resources from the east to the west can promote the realization of carbon neutrality and sustainable development.

**Keywords:** digital transformation; carbon neutrality; green and sustainable development; channel of computing resources from the east to the west



## 1. Introduction

Energy is the basis for the survival and development of modern society. Rapid economic growth is inseparable from energy consumption. According to statistics, more than 80% of carbon dioxide emissions come from energy activities, and solving energy problems is closely related to achieving the dual carbon goals. DT is an advanced transformation based on digital transformation and digital modernization, which continues to touch the company's core business and create new business models. Studying the application of DT in the field of carbon neutralization is of great significance for achieving the ideal goal of carbon neutralization and supporting green, low-carbon and high-quality economic and social development. In addition, people must reduce carbon dioxide emissions and optimize the energy structure of all mankind.

Many scholars have studied the relationship between carbon neutrality and sustainable development. Mallapatty Smriti investigated the role of renewable energy in achieving the goal of carbon dioxide neutralization, that is, to compensate for gas emissions through tree planting or carbon dioxide capture and underground storage [1]. In order to achieve carbon neutrality, Caineng Z.O.U. proposed seven implementation strategies to build a new energy pattern of "three small and one large" [2]. Khan Syed Abdul Rehman used fixed effect and random effect models to empirically test carbon-free economic growth and international tourism through advanced logistics infrastructure and renewable energy consumption in developed countries [3]. Chen Chi would need many different technologies to reduce the carbon emissions of existing processes and effectively use carbon dioxide as a chemical and fuel raw material [4]. Song Qing-Wen prepared valuable compounds through green catalysis to improve atomic economy and enhance the sustainability of chemical processes [5]. Zhao Yun believed that sustainable transportation requires carbon neutral fuel to overcome the battery range and charging limits, and the scale limit of biofuels [6]. McCollum David L believed that the significant reallocation of the portfolio required to change the energy system would not be initiated by the current national autonomous contribution package [7]. The above studies all described the role of carbon neutralization and sustainable development, but did not incorporate DT.

DT promotes green and sustainable development. George Gerard discussed how digital technology can help address the major challenges of climate change and promote sustainable development, proposed a research agenda, and raised new issues for entrepreneurship, business models and ecosystems, as well as new approaches to thinking about trust and institutional logic [8]. Sebhatu Samuel Petros understood how to use the UN sustainable development goals to guide stakeholders to participate in change, respond to global challenges, and control new business social practices driven by value based business models [9]. Elgohary Esam discussed the concept of sustainable development and how the communication and information technology industry can provide technical support for achieving sustainable development [10]. Savchenko A. B believed that the continuous implementation of the technological model capability of economic integration has created opportunities for significantly improving the productivity of urban infrastructure and industrial sectors, and for developing the knowledge and experience economy therein [11]. The above studies all described the role of DT, but there are still some deficiencies in sustainable development. It can be seen that DT plays a positive role in achieving carbon neutrality and sustainable development. It can promote enterprises' green technological innovation and reasonable planning, and effectively promote enterprises' digital transformation to meet the challenges brought by market development.

This paper first analyzes the current challenges faced by carbon neutrality and sustainable development, and the significance of achieving carbon neutrality and sustainable development, then studies the nature and mode of DT and the advantages of DT in helping achieve carbon neutrality and sustainable development, and finally analyzes the impact of DT on carbon neutrality and sustainable development, and constructs the implementation path of carbon neutrality and green sustainable development.

In order to achieve the goal of carbon neutrality and sustainable development, this paper analyzes the relative subordination degree and sustainable comprehensive evaluation level under the digital transformation by analyzing the entropy weight method, and then analyzes the risk transaction and digitalization degree of DT for enterprises by analyzing the resource utilization rate, energy consumption rate and low-carbon financing effect before and after the digital transformation. Through experimental analysis, it is found that DT can effectively improve the digitalization of enterprises and reduce transaction risks. Compared with other documents, this paper focuses on comparing the perfection of the carbon trading system and the effect of risk management in enterprises before and after DT, and also carries out relevant content analysis through comparative tests. In addition, this paper also analyzes the comprehensive indicators of sustainable development under the digital transformation by combining the entropy weight method.

## 2. Evaluation of Carbon Neutralization and Sustainable Development Factors

(1)   Current challenges to carbon neutrality and sustainable development

As shown in Figure 1, to achieve the long-term goals of carbon peak and carbon neutrality, there are problems that must be overcome; these include imperfect policy supervision mechanisms, a low degree of industry digitalization, and an imperfect carbon market and trade system. Figure 1 is mainly concerns the low degree of industry digitalization caused by imperfect policies and regulations, resulting in an imperfect carbon market trading system and imperfect environmental protection mechanism. These are the main problems hindering the development of carbon neutrality, and the realization of carbon neutrality must also solve these problems. First, the policy supervision mechanism is not perfect. At present, the policy framework and mechanism related to carbon peak and carbon neutralization are not perfect, and the guidelines and regulatory mechanism supporting both carbon peak and carbon neutralization are still being developed. After the $CO_2$ ceiling was included in the multisectoral work plan of the Economic Conference, many provinces also included the $CO_2$ ceiling in their local plans, but most regions have not yet formulated clear goals and regulatory measures. Although these policies strengthen the energy conservation standards and regulatory framework, the current carbon emission statistics, monitoring system and regional mechanism need to be further improved and perfected to classify and evaluate carbon emission reduction targets [12]. Therefore, existing directives cannot effectively control carbon peak and carbon neutralization. Second, the degree of industry digitalization is low. Traditional industries are still a part of the great development. The low level of digitalization and synergy in the industrial chain limits the overall efficiency of social resource allocation. Industrial restructuring, DT and modernization are some of the important ways to achieve $CO_2$ peak and neutralization. Therefore, people must speed up the digital and green upgrading process of traditional high energy consumption and low digital industries. Third, the carbon market and trading system are not perfect. From the perspective of the carbon market, the carbon market and carbon trading system are still in their initial stages. At present, there are some problems in the carbon emission trading market, such as inconsistent rules, difficulty in forming a market mechanism, opaque information, uncertain prices, and difficulty in collecting and verifying the historical wastewater data of enterprises. In terms of funds, there are further problems that need to be overcome, such as insufficient funds, insufficient channels and uncertain sources of funds. Fourth, the environmental protection system and institutional mechanism are not perfect. The support of the environmental protection system is insufficient, and the resource and environmental constraints related to economic growth are not clear, but all environmental protection systems and mechanisms are progressing slowly. Some institutional arrangements give the impression that environmental protection is insufficient to keep pace with economic growth. What is more important is that some systems are not implemented effectively in practice, which leads to the ineffective implementation of institutional arrangements regarding environmental protection that possess anti-environmental characteristics in the traditional development model, particularly in the environmental legal system. In addition, there is a lack of very effective environmental monitoring and regulatory rules to monitor and manage the environment, leading to unclear responsibilities, the ambiguous positioning and division of labor, and uncertain separation of tasks. With regard to law enforcement mechanisms, due to the impact of different interests, environmental monitoring and law enforcement need to be improved.

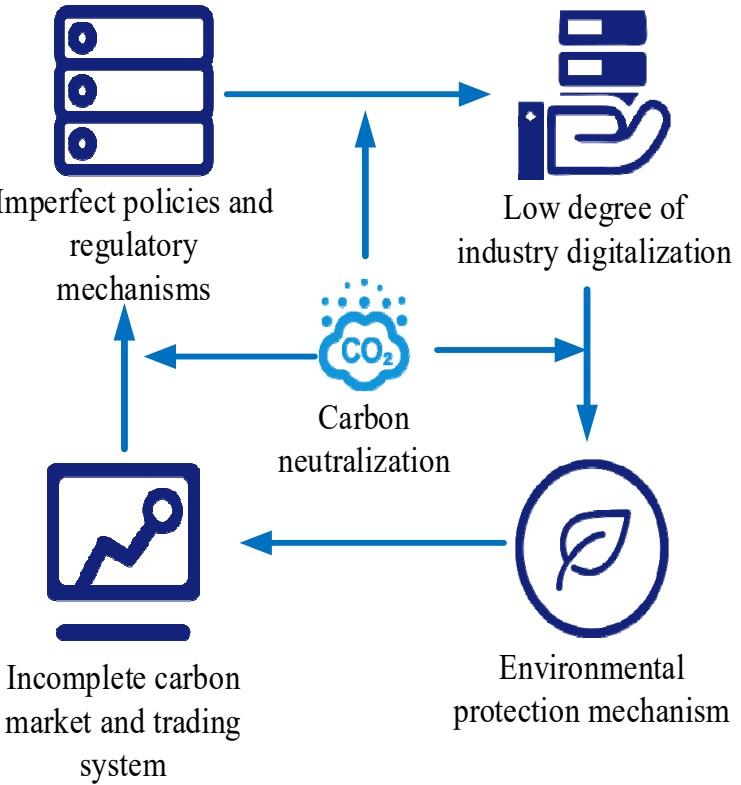

**Figure 1.** Challenges of carbon neutralization.

(2)    The road to carbon neutrality and sustainable development

Carbon neutralization means that the positive and negative emissions of carbon are balanced, and that carbon dioxide emissions are almost zero. It promotes urban energy supply, replaces fossil energy with clean energy, and completely separates economic growth and carbon dioxide emissions [13]. As shown in Figure 2, the realization of the carbon dioxide neutral target depends on energy conservation and emission reduction on the one hand, and on the increase in carbon emissions on the other hand. Carbon neutralization mainly relates to energy conservation and emission reduction, so as to meet the demand for carbon neutralization. In the long run, cities may be forced to promote the harmonious development of the economy and the environment, which is an inevitable choice for sustainable development and high-quality development. To mobilize the enthusiasm of all parties, carbon dioxide emission rights should be assessed through equity certificates. On the basis of the continuous promotion of renewable energy, a perfect carbon dioxide emission trading single market should be established to allocate carbon dioxide emission rights and resources, so as to achieve regional and industrial energy conservation, make full use of the differences in emission reduction costs, and encourage all stakeholders to emphasize the importance of reducing carbon dioxide emissions and using clean energy. The carbon trading market and carbon emission rights can be transferred to regions with a high carbon demand, reduce the proportion of energy-intensive industries, and optimize the energy structure. As the main equity certificate for the development of the digital energy industry, the carbon dioxide emission right conforms to the basic principles of developing a low-carbon and sustainable digital energy industry. Linking carbon dioxide emissions with economic benefits would help optimize the carbon emission structure of regions and industries.

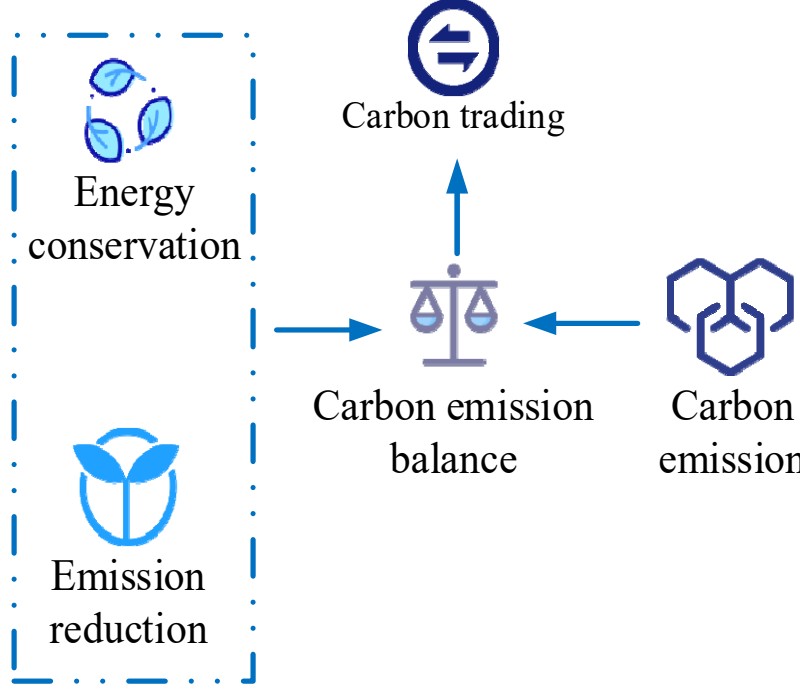

**Figure 2.** Objectives of carbon neutralization.

## 3. Evaluation of the Relationship between DT and Sustainable Development

(1)　Essence of DT

The main goal of DT is to improve efficiency and synergy, as shown in Figure 3. The core of DT is to apply digital awareness and digital technology to all aspects of production company management, and comprehensively and systematically change company activities, products, networks, management, methods and procedures, and tools. The core is the transformation of the development model, the redesign of the value system and the realization of the "four transformations". It changes the configuration mode and the dependence on the expansion path of traditional elements, namely, the dependence on capital, cost, labor, etc. Through resource reallocation and overall optimization, the project configuration model can be multi-point, global and dynamic optimization to improve enterprise efficiency. It also changes the business model, insisted on taking customers as the center, and jointly builds a business model with customers. From product management to customer management, it uses big data to accurately identify customer needs, effectively configure products, and quickly carry out marketing services [14]. In addition, people need to change the development path, open up new digital space, and realize digitalization. According to the requirements of sustainable development, people must improve the prospects for changing the development mode. The traditional development mode is the product of industrial civilization. Only, from the perspective of industrial civilization, it is difficult to realize the fundamental change of speed to really change the development mode. First, in the process of changing the development mode, ecological civilization and sustainable development can be realized. People should strengthen the concept of dynamic development, and make ecological civilization and sustainable development the fundamental point and direction of changing the development mode. This is the transition from traditional industrial civilization to ecological civilization, and from catching up development mode to sustainable development.

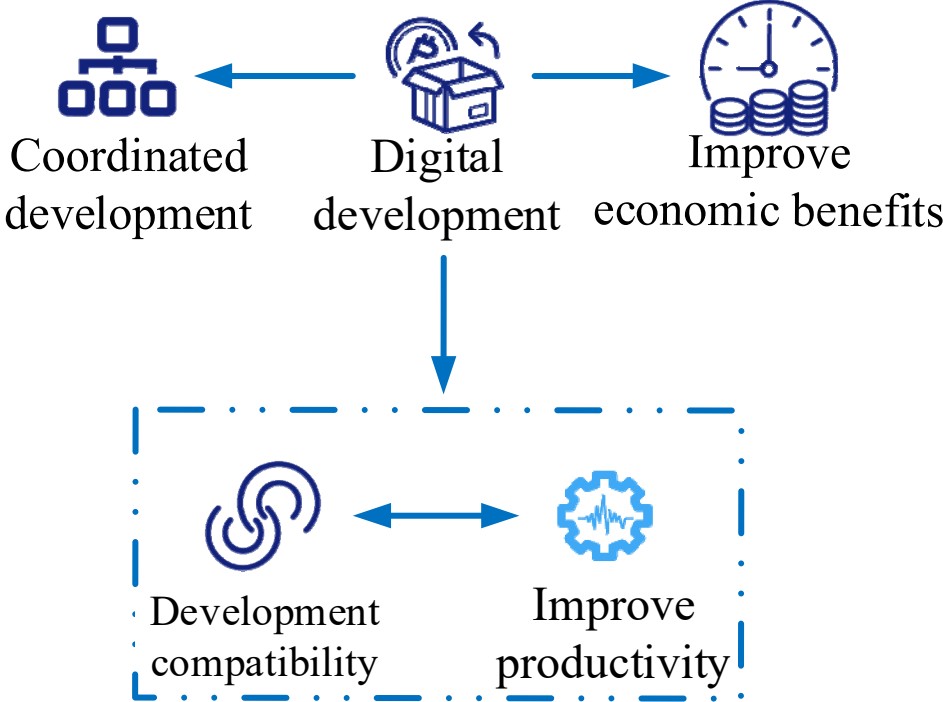

**Figure 3.** Objectives of digital transformation.

(2)    DT mode

Under the DT of the industry, transformation should be carried out in the following ways. The first is to consolidate the foundation and promote the construction of the platform. The potential of this platform is the core capability and foundation for the full integration of industrial and digital technologies, accelerating the transition by implementing data, applications, services and transactions for all users. Expanding the connection ratio and connection size is the default value of network connection. Facing the huge market of the Internet of Things, people must focus on popularizing the Internet of Things. The second is to establish an operating system. Corporate governance is a dynamic circular system. Through DT, it breaks down the weaknesses and bottlenecks of corporate governance and promotes the internal circulation of enterprises. It integrates cloud functions, improves the capability of independent platforms, and consolidates the foundation of digital unity. Data management and production planning are used to create an intelligent intermediate platform, build an intelligent operation brain based on data and system integration, and use big data technology to promote in-depth online integration. If the government wants to improve transparency, accountability and efficiency, DT can become a key driver of government change [15]. The third is to build a digital cooperation ecosystem. Creating an ideal digital ecology is the way to deeply integrate the digital economy with the real economy. People would continue to focus on digital households and key vertical industries, strengthen the close cooperation system based on key industry scenarios, promote the optimization and upgrading of the value chain, resource sharing and trade cooperation, and improve the industrial chain to achieve complementary advantages and mutually beneficial cooperation.

(3)    Advantages of DT for carbon neutralization and sustainable development

There are three main advantages to help carbon neutralization and sustainable development under DT. The first is to improve resource utilization. As a new production factor, DT is widely used in the field of carbon dioxide emission reduction [16]. Through this algorithm, the platform components can record consumer needs more accurately and timely, and improve the real-time interaction between production and consumption. At the same time, the platform can quickly identify the effective needs of a large number of buyers

and sellers, improve matching efficiency, and reduce resource utilization and consumption. From expansion to the reallocation of unused resources, it increases the production and reuse of existing storage resources and reduces the opportunity cost of production factors. The second is to reduce energy consumption. DT significantly improves industrial productivity and enterprise decision-making efficiency, and reduces industrial energy consumption. Under the pressure of carbon neutral indicators, DT is indispensable when aiming to accelerate the construction of new energy systems and various intelligent energy systems, and promote the green transformation of energy-intensive industries [17]. The use of big data and digital platforms can effectively improve energy production, transportation and consumption efficiency by accurately tracking the energy consumption and carbon dioxide emissions of construction companies and cities. The third is to solve low-carbon financing. DT is an important technical tool of green finance. Digital financing can improve the flexibility and convenience of services and improve the efficiency of financial services and capital distribution. Through the digital service platform, the sharing mechanism of resources, market, technology and financing information is established to promote the understanding of green low-carbon projects and enterprise supply chain. People would improve the integration of capital, the value chain and industrial chain, help the carbon trading market better play its role in resource allocation, risk management and market pricing, and promote the development of the carbon trading market.

## 4. Application of Entropy Weight Method in Carbon Neutralization and Green
*Sustainable Development*

In order to understand the specific effects of carbon neutralization and sustainable development under DT, this paper analyzes the judgment matrix and entropy weight vector of sustainable development after DT through the entropy weight method, and obtains the comprehensive relative membership degree and evaluation grade of sustainable development after DT according to the analysis results. The entropy weight method can use entropy value to judge the dispersion degree of sustainable development indicators, and also can judge the impact of digital transformation indicators on the comprehensive evaluation of sustainable development. First, the judgment matrix A of carbon neutral sustainable development is constructed as follows:

$$A = \left( a_{ij} \right)_{m \times n} \tag{1}$$

Among them, $a_{ij}$ is the judgment index, and then the judgment matrix is normalized to obtain the normalized judgment matrix B as follows:

$$B_{ij} = \frac{a_{ij} - a_{\min}}{a_{\max} - a_{\min}} \tag{2}$$

Among them, $a_{\max}, a_{\min}$ are the most satisfied and the least satisfied with the sustainable development of different regions under DT, and then the entropy of the evaluation index under DT is determined as follows:

$$C_i = -\frac{1}{\ln m} \left[ \sum_{j=1}^{n} x_{ij} \ln x_{ij} \right] \tag{3}$$

$$x_{ij} = \frac{B_{ij}}{\sum\limits_{j=1}^{m} B_{ij}} \tag{4}$$

Among them, m is the number of evaluation indicators, and $x_{ij}$ is the detection value of evaluation indicator entropy. Formula (4) is modified to obtain the modified detection value:

$$x_{ij} = \frac{B_{ij} + 1}{\sum\limits_{j=1}^{m} \left( B_{ij} + 1 \right)} \tag{5}$$

Then, according to the corrected detection value, the entropy weight vector of sustainable development can be obtained as follows:

$$r_i = \frac{1 - C_i}{n - \sum\limits_{j=1}^{n} C_i} \tag{6}$$

Then, according to the fuzzy recognition, the comprehensive relative membership of DT can be obtained as follows:

$$t = \frac{1}{1 + \left( \frac{d_1}{d_2} \right)^n} \tag{7}$$

Among them, $d_1$, $d_2$ are the maximum and minimum range value of the membership matrix, respectively, and the final evaluation grade of sustainable development can be obtained as follows:

$$G = \left( t / \sum\limits_{i=1}^{n} t_i \right) \times n \tag{8}$$

## 5. Calculating the Impact of DT on Carbon Neutralization and Sustainable
*Development from East to West*

(1) Background analysis of east and west calculation

Through the construction of new data centers, cloud computing and integrated big data network systems, the Eastern Digital and Western Computing will guide the computing demand from the east to the west, optimize the construction layout of data centers, and promote the cooperation between the east and the west. This will enable Western resources to more fully support Eastern data business and better promote digital development. In the eastern and western calculations, figures refer to data, and calculation refers to computing capacity, namely the data processing capacity of the United Nations [18]. The East Digital West Computing is a new data center that is building cloud computing and a big data-integrated computing network system, is orderly managing the computing needs of the east and west, is optimizing the construction layout of the data center, and is promoting the cooperation between the east and west. In the east and west, orderly and intensive data processing is carried out from east to west. By calculating the artery, we can alleviate the energy shortage in the east and open up a new development path for the west. The implementation of East–West Computing will help to improve the overall level of the computing capacity of the country by establishing a national integrated data center, expanding the scope of computing infrastructure, improving the efficiency of computing capacity utilization and realizing the intensive development of the country. It will also promote green development, expand the layout of the data center in the west, significantly improve the utilization rate of green energy, absorb green energy from the Near East, continuously optimize the energy efficiency of the data center through technological innovation, and formulate low-carbon measures from large to small [19]. By promoting the balanced development of the region, organizing the computing facilities in the east and west, realizing the efficient transfer of relevant industries, and promoting the data flow and value transfer between the east and west, it will expand the development area in the east and promote the formation of a new development model in the west.

(2)    The influence of the east counting and the west counting on carbon neutralization

With the establishment of a new computer network system combining a data center, cloud computing and big data, channeling more computing resources from the eastern areas to the less developed western regions would orderly manage the computing needs of the East and the West, optimize the layout of the data center, and promote harmony between the East and the West. In order to realize the common development of digitalization and greening of the information and communication technology industry, the new data center must establish a new infrastructure to combine various data sources, green low-carbon technologies and safe and reliable capabilities to provide efficient services [20]. At present, the eastern region has high requirements for computing capacity, but the climate, resources and environment of the eastern region are not conducive to the establishment of low-carbon and green data centers. The western region can use information technology infrastructure to give full play to its advantages in climate, energy, land and other fields. On the one hand, the development of connectivity between the East and the West gives full play to the advantages of climate, national and renewable energy and other resources in the West, in order to meet data needs and to achieve the low-carbon and green development of the data center. On the other hand, it can absorb green energy near the West, ease the power pressure and energy consumption indicators in the east, and optimize resource utilization. The implementation of the channels computing resources from the east to the west is the key to achieving a reasonable layout of data centers, optimizing supply and demand, strengthening green connectivity, improving the country's comprehensive computing capacity, and achieving large-scale and intensive computing capacity. Promoting green development, absorbing green energy from the western region, and constantly optimizing the energy efficiency of the data center would help promote data flow and value transfer between the East and the West, and expand the development space of the East. The implementation of the framework requires the joint efforts of several stakeholders. The government's improvements to the market mechanism would help speed up asset governance. Enterprises would strengthen cooperation with local governments and industry. The first is to develop a carbon neutral mechanism. By promoting the docking of their own green ecosystems, a top-down governance mechanism would be formed to promote green asset trading, renewal and intelligent management.

(3)    Calculating the implementation path of carbon neutralization and green sustainable development from the east to the west

The main path to achieve carbon neutrality and green sustainable development can be started from the following aspects, as shown in Figure 4.

(1)    Balanced planning layout

To achieve carbon neutral and balanced sustainable development, people must define the concept of value-sharing development at the national level, formulate a balanced development plan, and scientifically design the concept of development. The first is to give full play to the time–space advantages of digital economy, support the comparative advantages of the digital economy industry and regional development, and promote innovative development. The second is to unify the collection, allocation, application and sharing rules and standards of data resources, and promote the integration of data resources. At the same time, people must make full use of the huge Internet user data resources accumulated in the field of digital economy to create an overall design and conduct research, promote major technological breakthroughs, and gain new competitive advantages. Finally, people should continue to increase investment in research and the development of the digital economy and create a more favorable environment for development through the establishment of large special innovation funds.

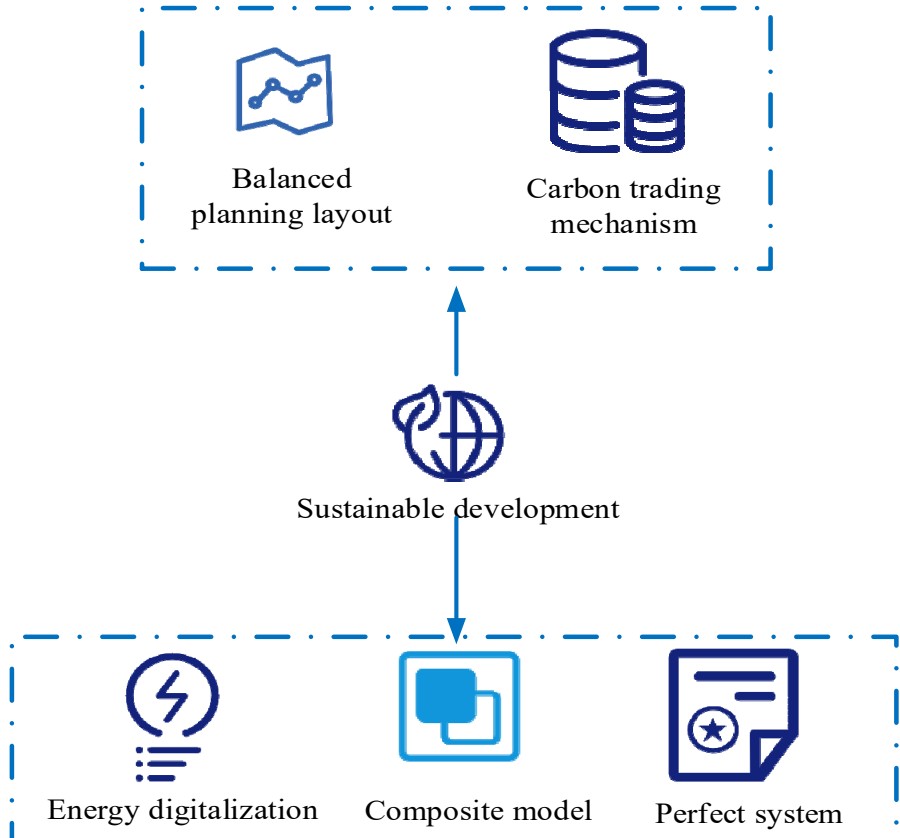

**Figure 4.** Implementation path of carbon neutralization and green sustainable development.

(2)    Promoting energy digitalization

People should strengthen the planning of the digital energy transformation demonstration area and take the lead in implementing digital clean energy projects. On the one hand, it provides financial and political support for clean energy and digital technology projects; on the other hand, it provides good human resources for the application of clean energy and digital technology. As for the approved projects, people would track and evaluate them strictly in accordance with the planning plan, accelerate the construction of a number of typical demonstration projects, transform the concept of industrial development through industrial demonstration, and promote the comprehensive digital and intensive development of the power industry. In the field of docking development, cloud computing, the Internet of Things, artificial intelligence and other intelligent technologies are used to improve the collection efficiency, expand the development side of docking capabilities, and realize the development process of precision, online resources and intelligence. In the field of energy consumption, smart technologies such as big data and artificial intelligence can change energy consumption, reduce the demand for fossil energy, and promote the formation of new energy consumption concepts.

(3)    Building a carbon trading mechanism

People must accelerate the establishment of and improvements in a national unified carbon trading platform, improve the carbon emission pricing mechanism, establish a unified carbon trading department, and formulate industrial laws and regulations. A carbon dioxide financing mechanism should be established, including carbon dioxide loans, futures and options. It strengthens the application of computer technology in the East and West, playing an important role in technological innovation and industrial transformation. With DT as the breakthrough point, people should independently innovate core technologies, increase investment, actively solve a number of key technologies, and accelerate the

development of digital technology and industrial innovation. Digital technology can be used to anonymously store digital records of the distributed production, consumption and circulation of renewable energy. Based on storage technology, enterprises can obtain personal credit points and build energy credit in a digital society, laying a foundation for the future carbon financial system and carbon neutral society.

(4)    Creating a centralized development element combination model

The road of harmonious development of ecological civilization and economy must follow the road of connotative development. The extensive development mode is an important factor of ecological destruction. It leads to the extensive utilization of resources and destructive development, leading to a "double surplus" of resources and the environment. In other words, it leads to the unconventional use of natural resources and unconventional emissions of pollutants. On the other hand, the extensive development model makes the industrial structure of different regions similar, resulting in huge resource waste and pollution, so it is necessary to establish an intensive development model. This means promoting the integration, concentration and intensive use of resource elements. Only in this way can people truly create a development model that saves and recovers resources, maximally saves and uses resources and minimizes the environmental impact.

(5)    Forming a sound institutional arrangement

The systematic management of resource protection and environmental protection is an important part of the development model. The basic mechanism for the traditional development model to ignore environmental and resource protection is to exclude relevant systems from the development model. The existing institutional allocation is not conducive to the protection of resources and the environment, and seeks a development model conducive to ecological civilization. In order to develop, people must incorporate institutional arrangements into the development model and form a logical structure for the new development model, which is conducive to resource compensation and financing for environmental protection. It includes the establishment of financial and credit systems, the establishment of a strict resource incentive and protection legal system, and the establishment of a market mechanism for resource conservation and environmental protection.

## 6. Experimental Evaluation of the Relationship between DT and Carbon
*Neutralization*

In order to study the impact relationship between DT and carbon neutralization, this paper has analyzed the significance of DT to the realization of the carbon neutralization goal by analyzing the resource utilization rate, energy consumption rate and low-carbon financing effect of channels computing resources from the east to the west; it compared and analyzed the carbon emission effect before and after DT, so as to analyze carbon neutralization and sustainable development. First, this paper investigated ten regions, and analyzed the average value of resource utilization, energy consumption and the low-carbon financing effect before and after DT. The specific mean comparison is shown in Table 1. These ten regions are Guangdong, Liaoning, Hubei, Shaanxi, Yunnan and Tianjin, Chongqing, Shenzhen, Xiamen, and Hangzhou. The reason for choosing these ten regions is that these regions are pilot cities for low-carbon development in the region, and there are many industrial enterprises that release many carbon emissions every day. These relevant data are found on the official website of the National Bureau of Statistics and the China Carbon Emissions Database. Therefore, studying the resource utilization rate, energy consumption rate and financing effect of these ten regions is of great help when aiming to analyze the impact of digital transformation on carbon neutrality and sustainability, and helps to verify the specific role of digital transformation.

**Table 1.** Resource utilization rate, energy consumption rate and low-carbon financing effect before and after digital transformation.

|  | Before Digital Transformation | After Digital Transformation |
|---|---|---|
| Resource utilization | 64.37% | 85.48% |
| Energy consumption rate | 75.46% | 32.57% |
| Low carbon financing effect | 48.56% | 74.95% |

According to the data described in Table 1, the resource utilization rate and low-carbon financing effect after DT were better than those before DT, while the energy consumption rate after DT was lower than that before DT. After DT, the resource utilization rate was 85.48%; the energy consumption rate was 32.57%; and the low-carbon financing effect was 74.95%. Before DT, the resource utilization rate was 64.37%; the energy consumption rate was 75.46%; and the low-carbon financing effect was 48.56%. It can be seen from the comparison that the resource utilization rate after DT was 32.8% higher than that before transformation; the energy consumption rate was 56.8% lower than that before transformation; and the low-carbon financing effect was 54.3% higher than that before transformation. After DT, all kinds of renewable energy were fully utilized, and the financing channels were more extensive, which not only improved the resource utilization rate, but also reduced the consumption rate of non-renewable resources.

Digital technology promotes the rapid decarbonization of energy, transportation and industry, promotes the common economic cycle, realizes dematerialization, effectively uses resources and energy, monitors and protects ecosystems, and helps protect global public resources. This is a sustainable technology. The digital revolution is not only a "tool" to cope with the challenges of sustainable development, but is also a fundamental factor that leads to unstable changes in the world and is an important driving force for social change. Digitalization is an important driving force for the decarbonization of energy and transportation systems, the circular economy and social transformation. This may lead to energy maximization, sustainable urban reconstruction, ecosystem protection and monitoring, etc. The threats and opportunities of the modern digital revolution must be included in the evolution of the concept of sustainable development in order to change the description of the concept of sustainable development itself. Digital technology and its integration will certainly improve humanity's physical and cognitive abilities. Of course, improving cognitive ability is the main problem. The Internet and mobile applications provide important enhancement technologies for human cognitive ability, external memory and knowledge accumulation.

This paper then analyzed the carbon emission effect before and after DT, and the specific changes are shown in Figure 5.

According to the histogram depicted in Figure 5, the carbon emission effect before DT gradually increased over time, while the carbon emission effect after DT continuously decreased over time. The average value of the carbon emission effect before DT was about 0.71, and the average value of the carbon emission effect after DT was about 0.37. On the whole, the initial value of the carbon emission effect before DT was 0.51, which increased to 0.92 in the seventh year, and increased by 0.41 in the whole process; the initial value of the carbon emission effect after DT was 0.52, which decreased to 0.20 in the seventh year, and decreased by 0.32 in the whole process. It can be seen from the comparison that the carbon emission effect after DT was 47.9% lower than that before DT. Enterprises after DT started to use green energy for production, and their carbon emissions naturally decreased. Then, the entropy weight vector and evaluation grade of sustainable development after DT were analyzed by using entropy weight method. The specific changes are shown in Figure 6.

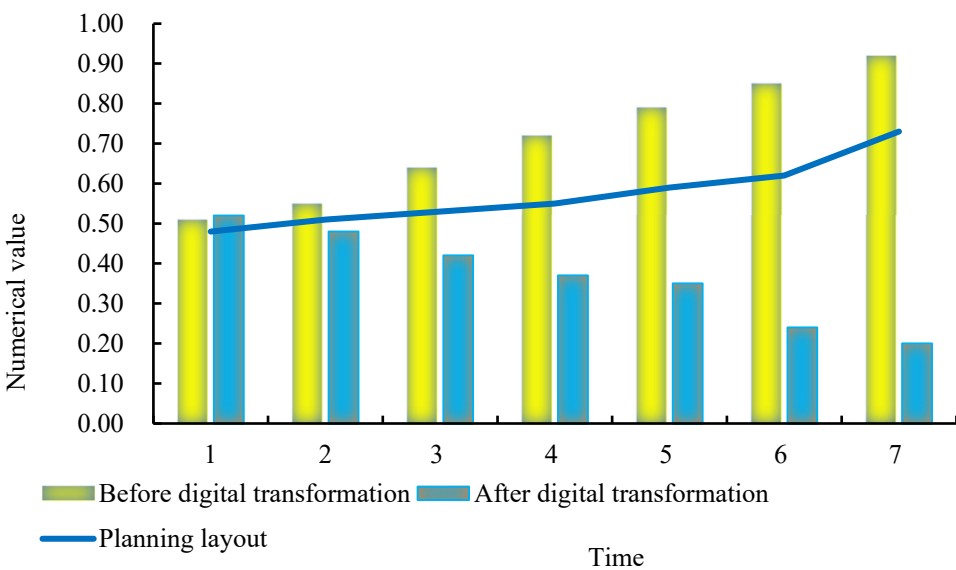

**Figure 5.** Carbon emission effect and planning layout before and after DT.

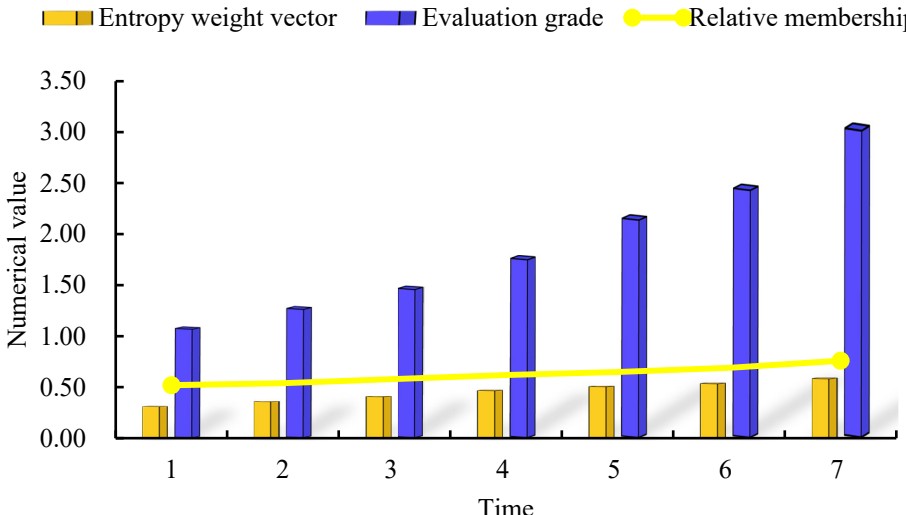

**Figure 6.** Entropy weight vector and evaluation grade of sustainable development after digital transformation.

According to the curve depicted in Figure 6, the entropy weight vector and evaluation grade of sustainable development after DT have been rising over time. The average value of the entropy weight vector of sustainable development was about 0.47, and the average value of the evaluation grade was about 1.93. On the whole, the initial value of entropy weight vector was 0.32, which increased to 0.60 on the seventh day, with an increase of 0.28 in the whole process; the initial value of the evaluation grade was 1.10, which rose to 3.10 in the seventh year, and rose by 2.00 in the whole process. The initial value of relative membership was 0.52, which rose to 0.76 in the seventh year, and rose by 0.24 in the whole process. The increase in the entropy weight vector and evaluation grade proved that the development after DT correlated with green and sustainable development, and in the context of channeling computing resources from the east to the west, renewable energy and green technology could therefore be integrated into social development. Later, this paper analyzed the degree of industry digitalization and the perfection of the carbon trading system under DT, and compared it with the industry digitalization and carbon trading system before DT, as shown in Figure 7.

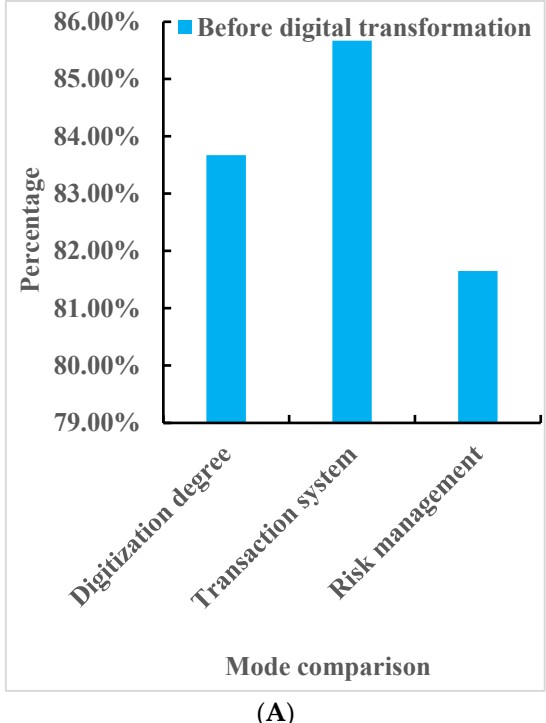
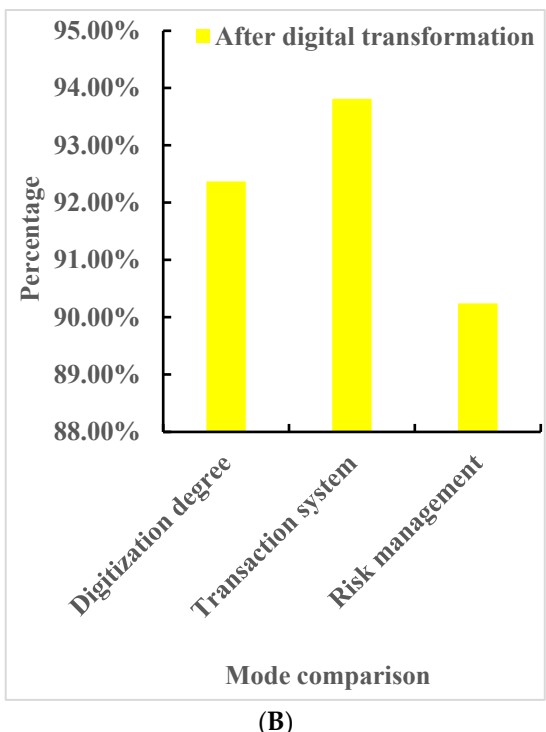

**Figure 7.** Industry digitalization and carbon trading system perfection under digital transformation. (**A**): Before digital transformation, (**B**): After digital transformation.

Figure 7A shows the data before the experiment, and Figure 7B shows the data after the experiment. According to the comparison between Figure 7A,B, the post-DT industry digitalization, the carbon trading system perfection and the risk management effect were better than those before the DT. The post-DT industry digitalization was 10.4% higher than that before the DT, and the carbon trading system perfection was 9.5% higher than that before the DT. The effect of risk management was 10.5% higher than that before DT. The carbon trading system in the channel of computing resources from the east to the west was more perfect, which standardized the behavior of carbon trading and the carbon emission market to a certain extent, and also promoted the transformation of enterprises using renewable energy. It not only achieved the goal of carbon neutrality, but could also can promote the sustainable development of enterprises and even society. Finally, we will analyze the matching rationality, production efficiency and decision-making efficiency of enterprises under DT, and compare with the relevant data before and after DT. We have investigated the changes in three enterprises before and after DT, and taken the average value. The specific comparison is shown in Figure 8.

Figure 8 shows that the matching rationality, production efficiency and decision-making efficiency of enterprises after DT are higher than those before DT. Before DT, the matching rationality of enterprises was 68%, the production efficiency was 75%, and the decision-making efficiency was 73%. After DT, the matching rationality of enterprises is 88%, the production efficiency is 86%, and the decision-making efficiency is 84%. The comparison shows that the matching rationality of enterprises after DT is 20% higher than that before DT, the production efficiency is 11% higher than that before DT, and the decision-making efficiency is 11% higher than that before DT. It can be seen that the matching rationality, production efficiency and decision-making efficiency of enterprises have been greatly improved after DT.

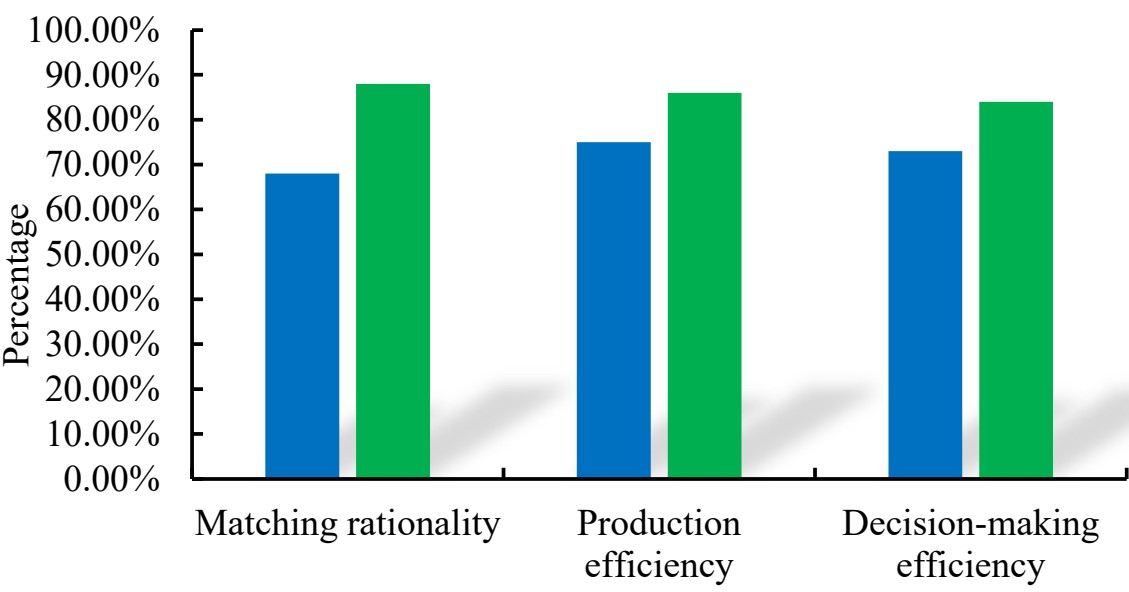

Index

**Figure 8.** Changes in matching rationality, production efficiency and decision-making efficiency of enterprises before and after DT.

Through the above experimental analysis, it can be seen that only by carrying out digital transformation can enterprises promote the realization of the goal of carbon neutrality and achieve sustainable development. Moreover, digital transformation can also reduce the transaction risk of enterprises, improve the utilization of resources and enhance the construction process of digital value. In addition, the government should also increase investment in carbon neutrality to promote regional renewable energy consumption, reduce carbon emissions, and achieve the goal of sustainable development. It should also improve institutional security to promote improvements in the carbon trading mechanism.

## 7. Conclusions

Channeling computing resources from the east to the west provides a visible and reliable regulatory environment for carbon peaking and carbon neutralization, stimulates DT in traditional industries, and plays an active role in carbon peaking and carbon neutralization. It has optimized business processes, reduced operating costs, improved collaboration efficiency, established a reliable and efficient carbon trading platform and market, and promoted carbon peak and carbon neutralization. Governments and enterprises can promote carbon peak and carbon neutrality by optimizing business processes, reducing operating costs, improving collaboration efficiency, establishing reliable and efficient carbon trading platforms and markets. Moreover, digital transformation can effectively improve the resource utilization of enterprises and reduce energy consumption. If people want to continue to use clean energy in the future and promote the comprehensive implementation of channeling computing resources from the east to the west, people must increase comprehensive efforts, go all out, enhance the influence and cohesion of the computing center, and promote computing integration and coordinated development. As an important means to promote green energy conservation and achieve the goal of carbon neutrality, the channeling of more computing resources from the eastern areas to the less developed

western regions would give these areas access to the climate, energy and environmental benefits of the western region. This is conducive to completing the development of the resource-rich western data center, increasing the supply of renewable energy, promoting the consumption of renewable energy in surrounding areas, strengthening data, computing capacity and energy synergy, and achieving carbon peak and carbon neutral targets.

**Author Contributions:** Conceptualization, Z.W. and X.W.; methodology, Z.W.; software, X.W.; validation, Z.W., X.Z. and H.H.; formal analysis, J.Y.L.; investigation, H.H.; resources, Z.W.; data curation, X.W.; writing—original draft preparation, Z.W. and X.W.; writing—review and editing, J.Y.L.; visualization, X.W.; supervision, X.Z.; project administration, J.Y.L.; funding acquisition, X.W. All authors have read and agreed to the published version of the manuscript.

**Funding:** This work was supported by the project of National Social Science Foundation in 2022 with the project number 22BMZ017.

**Institutional Review Board Statement:** Not applicable.

**Informed Consent Statement:** Not applicable.

**Data Availability Statement:** The data that support the findings of this study are available from the corresponding author upon reasonable request.

**Conflicts of Interest:** These are no potential competing interests in our paper. All authors have read and agreed to the submitted manuscript. We confirm that the content of the manuscript has not been published or submitted for publication elsewhere.

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
