# Peer review of "Evaluation of Digital Transformation to Support Carbon Neutralization and Green Sustainable Development Based on the Vision of “Channel Computing Resources from the East to the West”"

_sustainability, doi:10.3390/su15076299_

Round 1
Reviewer 1 Report
I suggest that the article be edited by a native speaker of English, as there are many points of style (such as names written with the last name first instead of the first name first) that are inconsistent with standards.
Additionally, the authors could provide more context on the "East to the West" paradigm.
Author Response
I suggest that the article be edited by a native speaker of English, as there are many points of style (such as names written with the last name first instead of the first name first) that are inconsistent with standards.
Additionally, the authors could provide more context on the "East to the West" paradigm.
Answer: I have added the background of East and West Counting to the text
Reviewer 2 Report
Please see the attachment.

Author Response
Evaluation of Digital Transformation to Support Carbon Neutralization and Green Sustainable Development Based on the Vision of “Channel Computing Resources from the East to the West”
Comments: major revision.
Digitization has brought profound changes to the economic and social development model, and the pattern of urban competition. This paper analyzed the significance of carbon neutralization and the challenges faced by sustainable development to study the advantages of carbon neutralization under Digital Transformation (abbreviated as DT), and finally proposed the implementation path of carbon neutralization and sustainable development based on the channel computing resources from the east to the west. The authors present an interesting topic and have done a lot of work on this topic, but due to the following comments, I suggest a major revision.
1- The object of the article research is unclear. It should be pointed out in the tittle. Why selected this country, its justification should be added in the main text.
Answer: In response to your question, I have added the research object to the experimental part.
2- The literature review in the introduction highlights the shortcomings of previous studies. Among them, the second paragraph concludes by stating that "The above studies all described the role of carbon neutralization and sustainable development, but did not combine DT", and the third paragraph concludes by stating that "The above studies all described the role of DT, but there are still some deficiencies in sustainable development". Based on these shortcomings, this article needs to specifically point out what innovations this article has based on the previous literature in order to better reflect the innovation points of this article.
Answer: In response to your question, I have added innovations to the article.
3- Each part of the article introduces the relationship between DT, carbon neutrality and sustainable development in modules, but the connectivity of each part is not strong, making it difficult for readers to grasp the overall logic of the article. Therefore, it is suggested to add the introduction of the overall thinking of the article in the introduction to better reflect the logic of the article.
Answer: In response to your question, I have added research ideas to the article.
4- The subtitle of the second content of the second part of this article is " Significance of carbon neutralization and sustainable development", but there is no relevant content of the importance of carbon neutralization and sustainable development, which is clarified in the later explanation. Instead, it focuses on the relevant content of the carbon trading market and carbon trading system, with little correlation between title and content. It is suggested that the title be changed to "Pathways to carbon neutrality and sustainable development".
Answer: I made changes in the article
5- In the third content of the third part of the article, the article mentions that "At the same time, the platform can quickly identify the effective needs of a large number of buyers and sellers, improve matching efficiency, can reduce resource utilization and consumption". Here, it is also necessary to provide data that can prove the rationality and success rate of platform matching, so as to prove that platform matching can improve the efficiency of resource utilization.
Answer: I have added relevant experiments to the article.
6- In the third content of the third part of the article, the article mentions that "DT significantly improves industrial productivity and enterprise decision efficiency". Here, the author should provide specific data or mechanism for DT to improve industrial productivity and enterprise decision-making efficiency to prove that DT really improves industrial productivity and enterprise decision-making efficiency.
Answer: I have added relevant experiments to the article.
7- In the fourth part of the article, the entropy weight method is mentioned, but the method and its applicability are not properly explained. Therefore, the method should be briefly explained. The introduction can also include related content of the research method to highlight the methodological nature of the article.
Answer: I have added the nature of entropy weight method and research methods to the article.
8- In the fourth part of the article, the formula is right-aligned. It would be more appropriate to change it here to centre-aligned.
Answer: In response to your question, I have modified the formula part.
9- From the sixth part of the article, the author has summarised a number of data from ten regions, but has not explained in the article which regions were selected and what the sources of the data were, which should be clarified in the article.
Answer: I have added relevant areas to the article
10- The scale of the vertical coordinate used in the left and right figures in Figure 7 is inconsistent, which makes it difficult for the reader of the article to intuitively feel the difference between the two figures from the comparison. It is suggested that the two figures be combined into one or that the same vertical coordinate axis be used to make the data more intuitive.
Answer: I have modified the coordinate axis on the way.
11- The conclusion can be made more concrete by using bullet points.
Answer: I have refined the conclusion.
12- The article draws conclusions based on the available literature and appropriate methods, but does not make policy recommendations. Carbon neutrality and sustainable development, as a practical topic, should provide development suggestions based on relevant conclusions. The introduction may also include relevant research content to emphasize the practical nature of the article.
Answer: I have added research ideas and policy suggestions to the article.
Reviewer 3 Report
This study discusses the advantages of carbon neutrality under the digital transformation, and puts forward the implementation path of carbon neutrality and sustainable development based on the channel computing resources from the east to the west. The manuscript requires to address the following issues to make it more accessible to the reader. Specifically,
1) The author should provide a clear definition of DT in the first paragraph of the Introduction, since this conception is quite important of this article. Readers, even non-professional readers, may expect to understand the research object of the article at the beginning.
2) The literature review part is a simple list of research results, but lacks analysis and summary. The introduction does not highlight the importance of DT research in this field.
3) The research method is not clear enough. The composition of the article does not clearly show what and how research method is used. It is difficult to distinguish whether this article is a review or quantitative research.
4) The figures need to be revised to be more scientific. The text in some figures is missing, for example, the title in Figure 7A.
5) The innovation of the research is not clear.
Author Response
This study discusses the advantages of carbon neutrality under the digital transformation, and puts forward the implementation path of carbon neutrality and sustainable development based on the channel computing resources from the east to the west. The manuscript requires to address the following issues to make it more accessible to the reader. Specifically,
1) The author should provide a clear definition of DT in the first paragraph of the Introduction, since this conception is quite important of this article. Readers, even non-professional readers, may expect to understand the research object of the article at the beginning.
Answer: I have added the definition of DT to the text.
2) The literature review part is a simple list of research results, but lacks analysis and summary. The introduction does not highlight the importance of DT research in this field.
Answer: I have modified the text.
3) The research method is not clear enough. The composition of the article does not clearly show what and how research method is used. It is difficult to distinguish whether this article is a review or quantitative research.
Answer: I have modified the text.
4) The figures need to be revised to be more scientific. The text in some figures is missing, for example, the title in Figure 7A.
Answer: I have modified the text.
5) The innovation of the research is not clear.
Answer: I have modified the text.
Reviewer 4 Report
Dear Authors,
Thank You for so interesting and relevant Article entitled "Evaluation of Digital Transformation to Support Carbon Neutralization and Green Sustainable Development Based on the Vision of “Channel Computing Resources from the East to the West”.
1. The article is written on a relevant topic, well-structured, and logically proven.
2. The topic is highly original and relevant in the field.
3. The authors duly describe the subject area compared with other published material.
4. The authors could consider some minor improvements in the logic of the presentation of the results. I'd suggest substantiating the loop view of the logic of the presentation Challenges of carbon neutralization in Figure 1. Then I'd suggest describing the difference between Figures 1 and 2. Figure 2 "Objectives of carbon neutralization" doesn't look like a loop, as in Figure 1, but contains some similar elements.
5. The conclusions are consistent with the evidence and arguments presented. I’d recommend the authors add some paragraphs about the possibility of implementing digital tools for Sustainable development in section 6. Discussion (the topic supposes to consider the Sustainable development concept).
6. The references are appropriate, but I'd recommend extending the list of references.
7. Please disclose the abbreviations that first appeared in the text.
Author Response
Dear Authors,
Thank You for so interesting and relevant Article entitled "Evaluation of Digital Transformation to Support Carbon Neutralization and Green Sustainable Development Based on the Vision of “Channel Computing Resources from the East to the West”.
- The article is written on a relevant topic, well-structured, and logically proven.
- The topic is highly original and relevant in the field.
- The authors duly describe the subject area compared with other published material.
- The authors could consider some minor improvements in the logic of the presentation of the results. I'd suggest substantiating the loop view of the logic of the presentation Challenges of carbon neutralization in Figure 1. Then I'd suggest describing the difference between Figures 1 and 2. Figure 2 "Objectives of carbon neutralization" doesn't look like a loop, as in Figure 1, but contains some similar elements.
Answer: In response to your question, I have supplemented the description in Figure 1 and Figure 2.
- The conclusions are consistent with the evidence and arguments presented. I’d recommend the authors add some paragraphs about the possibility of implementing digital tools for Sustainable development in section 6. Discussion (the topic supposes to consider the Sustainable development concept).
Answer: For your question, I have added specific discussion to the article.
- The references are appropriate, but I'd recommend extending the list of references.
Answer: I have added corresponding references to the article.
- Please disclose the abbreviations that first appeared in the text.
Answer: In response to your question, I have added the abbreviation to the text.
Round 2
Reviewer 2 Report
The detailed response to my last report is satisfactory, I thus recommend acceptance of the revised manuscript.
Author Response
The detailed response to my last report is satisfactory, I thus recommend acceptance of the revised manuscript.
Answer: Thanks so much.
Reviewer 3 Report
Thank you for your second submission. Unfortunately, the revised part did not answer/improve the problems in the previous version very well. I suggest that the author make extensive modifications to the research, and also modify the drawing method of figures to make the article more scientific. In addition, pay attention to errors in format and writing.
Author Response
Thank you for your second submission. Unfortunately, the revised part did not answer/improve the problems in the previous version very well. I suggest that the author make extensive modifications to the research, and also modify the drawing method of figures to make the article more scientific. In addition, pay attention to errors in format and writing.
Answer: I have made corresponding modifications and marked them in the text.